# Cascaded Teaching Transformers with Data Reweighting for Long Sequence Time-series Forecasting

## Abstract

The Transformer-based models have shown superior performance in the long sequence time-series forecasting problem. The sparsity assumption on self-attention dot-product reveals that not all inputs are equally significant for Transformers. Instead of implicitly utilizing weighted time-series, we build a new learning framework by cascaded teaching Transformers to reweight samples. We formulate the framework as a multi-level optimization and design three different dataset-weight generators. We perform extensive experiments on five datasets, which shows that our proposed method could significantly outperform the SOTA Transformers.

## 1 Introduction

The *long sequence time-series forecasting* (LSTF) has drawn particular attention with Transformer-based models, like electricity prediction Li et al. (2019), financial predictions Zhang et al. (2022), and weather predictions Pan et al. (2022). The pairwise self-attention allows time-series points to directly attend to each other, which benefits the future forecasting from the historical observations. The pairwise computation's sparsity assumption Child et al. (2019) reveals that not every sample is equally prior. If we drop unnecessary pairwise connections without violating problem settings, we can acquire a stronger model with better generalization. For example, (*i*) Reformer Kitaev et al. (2020) uses the hashing bucket to select important query-key pairs (*ii*) LogSparse Transformer Li et al. (2019) only calculates attention pairs lying on the log-size step away from the diagonal. The sparsity assumption could be roughly considered as manipulating the weights within time-series, or more specifically, internal reweighting.

A key technical challenge preventing us from further performance improvement that solely relies on internal reweighting is the unavailability of a *specific design* of time-series weights under the sparsity-oriented framework. Instead, we can perform the reweighting explicitly, that is, reweight the whole train dataset so that the outlier samples can be excluded from the training. Meanwhile, we assign larger dataset-weights to those samples belonging to the main patterns of the dataset. Inspired by the knowledge distillation Hinton et al. (2015) and teacher-student learning Li et al. (2014), we can *teach* the Transformer to learn to weighted time-series. One common method is using pseudo labels Pham et al. (2021) to allow student learning from teacher outputs. If we cannot assign accurate labels, the student hardly learns as well as the teacher does, a phenomenon known as confirmation bias. To alleviate this drawback, we propose to reweight the inputs in a soft manner.

In this paper, we design a cascaded teaching framework. There is a sequence of Transformer models, where model $i$ teaches model $i + 1$. Specifically, model $i$ generates a pseudo time-series dataset, which is used to train model $i + 1$. Only the first teacher model uses the reweighted dataset, whereas other models use the pseudo time-series dataset generated by the previous model. Finally, the teacher model will update its dataset-weights based on the performance of student models. The experimental results show that we can significantly improve the performance of a time-series prediction model by simply reweighting the dataset while maintaining the sparsity assumption at the same time. More importantly, this cascaded learning framework is easily generalized to other seq-to-seq models.

The contributions are summarized as follows:

- We propose a cascaded teaching framework to reweight the teacher model's dataset based on the evaluation of student models, which forces the teacher model to get rid of noise data by generating proper dataset-weights.
- We design three dataset-weight generators to compress the trainable dataset-weights parameters.
- Extensive experiments on three datasets (five cases) have demonstrated its improvement in time-series learning.

## 2 RELATED WORK

### 2.1 TEACHER-STUDENT LEARNING

Teacher-student learning has been investigated in knowledge distillation Hinton et al. (2015), adversarial robustnessCarlini & Wagner (2017), self-supervised learningXie et al. (2020), etc. Most of these methods are based on pseudo-labeling. In these existing methods, the focus is to learn a student model with the help of a trained and fixed teacher model. In these works, the teacher model is not updated. In contrast, our method focuses on learning a teacher model by letting it teach a student model. The teacher model constantly updates itself based on the teaching outcome. Teacher-student learning has been investigated in several neural architecture search works as well Li et al. (2020); Trofimov et al. (2021); Gu & Tresp (2020). In these works, when searching the architecture of a student model, pseudo-labels generated by a trained teacher model whose architecture is fixed are leveraged. Our work differs from these works in that we focus on searching the dataset-architecture of a teacher model by letting it teach a student model where the student's dataset-architecture is fixed, whereas the existing works focus on searching the architecture of a student model by letting it be taught by a teacher where the teacher's architecture is fixed. In a recent work Pham et al. (2021) which was conducted independently of and in parallel to our work, the teacher model is updated based on student's performance. Our work differs from this one in that our work is based on a three-level optimization framework which searches teacher's architecture by minimizing student's validation loss and trains teacher's network parameters before using teacher to generate pseudo-labels, whereas in Pham et al. (2021) framework is based on two-level optimization which has no architecture search and does not train the teacher before using it to perform pseudo-labeling. In Such et al. (2019), a meta-learning method is developed to learn a deep generative model which generates synthetic labeled data. A student model leverages the synthesized data to search its architecture. Our work differs from this method in that we focus on searching the teacher's architecture via three-level optimization, while Such et al. (2019) focuses on searching the student's architecture via meta-learning.

### 2.2 REWEIGHTING TIME-SERIES

Reweighting is a technique widely used in the field of time series forecasting, but the objects of weighting are diverse: features Zhao et al. (2018), fuzzy relationship Yu (2005), gradient Zhang et al. (2019) etc. In addition to time series forecasting, reweighting is also used to periodic pattern mining Chanda et al. (2017), fault diagnosis Lv et al. (2017) and time-series classification Sellami & Hwang (2019). In these works, only one model is trained to generate weights on the corresponding components of the network. The difference between our method and the above works is that the dataset-weights trained by our framework are not coupled with the model, so they can be applied to other models trained on the same dataset. The weights for time-series can also be extracted with a Bayesian non-parametric way Saad & Mansinghka (2018) or even from the dataset dynamics Zhang et al. (2021). Our framework differs from these works in that our model reweights the entire training dataset by introducing teacher-student learning.

## 3 TEACHER-STUDENT LEARNING FRAMEWORK

**Notions** We start with a teacher model with parameters $T$ and a student model with parameters $S$. We feed the teacher model with training dataset $D_t^{(\text{trn})} = \{(s_i, t_i)\}_{i=1}^N$, where $s_i$ denotes the $i$-th input serie and $t_i$ is the corresponding outputs. Following the multi-task learning paradigm Caruana (1997), we assign different **dataset-weights** $p_i \in [0, 1]$ for each input samples $(s_i, t_i)$, which forms the dataset-weights $P = \{p_i\}_{i=1}^N$. In order to avoid introducing prior knowledge, we initialize

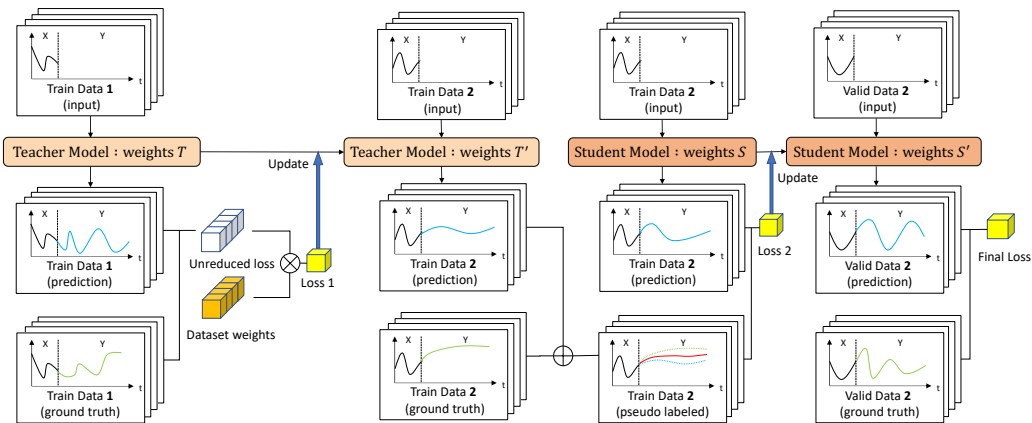

Figure 1: Cascaded Teaching Framework. Left: The teacher model performs one-step gradient descent with weighted loss. Middle: We use the teacher model with updated parameters T' to generate pseudo labels and mix them with ground truth. Right: The student models perform one-step gradient descent on the pseudo-labeled dataset. Then we use the validation loss of the student models to update the dataset-weights.

$P(A)$ to 1, so that all samples are treated equally in the beginning. Here we want to emphasize that the reweighting objective $(s_i, t_i)$ is not a single time point, but a complete time-serie sample which contains several time points, depending on the input/ouput length setting. Therefore, the dataset weights have no effect on the multi-head attention mechanism.

Without loss of generality, we assume the end task is time-series prediction. Our proposed cascaded teaching strategy could be divided into three stages. Firstly, we train a Transformer as the teacher model $T$ from the reweighted time-series. The training process could be formulated as:

$$T^*(A) \stackrel{def}{=} \min_T L(T, P(A), D_t^{(\mathrm{trn})}) = \min_T \sum_{i=1}^N p_i(A) l(T, s_i, t_i) \tag{1}$$

where the dataset-weights $P(A) = \{p_i(A)\}_{i=1}^N \in [0, 1]^N$ are generated by a group of parameters $A = \{a_j\}_{j=1}^{L_A} \in \mathbb{R}^{L_A}$, where $L_A$ represents the length of dataset-weights parameters A. If $p_i$ is close to 0, which means that $(s_i, t_i)$ is not important, the corresponding loss $p_i(A) l(T, s_i, t_i)$ is reduced to 0, which means that $(s_i, t_i)$ is excluded from training like holds a sparsity assumption. The parameter $A$ is fixed during this step. Otherwise, a trivial solution will be yielded where $P$ are all set to zero. The optimally trained $T^*(A)$ is a function of $A$ since they are functions of the reweighted loss, which is also a function of the parameters $A$.

Secondly, we use the previously learned $T^*(A)$ to generate a pseudo time-series dataset. We use another time-series dataset $D^{(\mathrm{unl})} = \{s_i\}_{i=1}^{L_u}$ without labels, where $L_u$ represents the length of $D^{(\mathrm{unl})}$ . For each $s_i \in D^{(\mathrm{unl})}$, we use $T^*(A)$ to predict the possible outputs $\hat{t}_i$. Then we get a pseudo-labeled time-series dataset $D^{(\mathrm{pse})}(T^*(A)) = \{(s_i, \hat{t}_i)\}_{i=1}^{L_p}$, where $L_p$ represents the length of $D^{(\mathrm{pse})}$. We use this pseudo-labeled time-series dataset to train another Transformer as student model $S$. The training process is minimizing the following loss:

$$S^*(T^*(A)) = \min_S [\gamma L(S, D_s^{(\mathrm{trn})}) + (1 - \gamma) L(S, D^{(\mathrm{pse})}(T^*)] \qquad , \tag{2}$$

where $\gamma \in [0, 1]$ denotes the self-study rate. If $\gamma$ is close to 1, then student model $S$ will study mainly on its own without consulting the "handouts" $D^{(\mathrm{pse})}(T^*)$ prepared by its teacher. Here we can use the input of $D_s^{(\mathrm{trn})}$ as the unlabeled dataset E for simplicity. The optimal parameters $S^*(T^*(A))$ are functions of $T^*(A)$.

Thirdly, we apply $S^*(T^*(A))$ to validate on $D^{\text{(val)}}$. We update $A$ by minimizing the validation loss. Putting all the pieces together, we get the following optimization framework:

$$
\begin{aligned}
\min_A \quad & L\left(S^*(T^*(A)), D^{\text{(val)}}\right) \\
s.t. \quad & S^*(T^*(A)) = \min_S[\gamma L(S, D_s^{\text{(trn)}}) + (1-\gamma)L(S, D^{\text{(pse)}}(T^*))] \\
& T^*(A) = \min_T L(T, P(A), D_t^{\text{(trn)}})
\end{aligned}
\tag{3}
$$

In this framework, there are three optimization problems. From bottom to up, the three optimization problems correspond to learning stage 1, 2, and 3 respectively. The first two optimization problems are nested on the constraint of the third optimization problem. These three stages are conducted end-to-end in this unified framework. The solution $T^*(A)$ obtained in the first stage is used to create a pseudo time-series dataset in the second stage. The time-series prediction model trained in the second stage is used to make predictions in the third stage. The parameters of the dataset-weights generator $A$ updated in the third stage changes the training loss in the first stage and consequently changes the solution $T^*(A)$, which subsequently changes $S^*(T^*(A))$.

## 3.1 OPTIMIZATION ALGORITHM

In this section, we develop a gradient-based optimization algorithm to solve the problem defined in Eq.(3). We approximate $T^*(A)$ using one-step gradient descend w.r.t $L(T, P(A), D_t^{\text{(trn)}})$ :

$$
T^*(A) \approx T' = T - \eta_t \nabla_T L(T, P(A), D_t^{\text{(trn)}})
\tag{4}
$$

Then we plug $T'$ into $L(S, D^{\text{(pse)}}(T^*))$ and get an approximated objective. We approximate $S^*(T^*(A))$ using one-step gradient descent w.r.t the approximated objective:

$$
S^*(T^*(A)) \approx S' = S - \eta_s\gamma\nabla_S L(S, D_s^{\text{(trn)}}) - \eta_s(1-\gamma)\nabla_S L(S, D^{\text{(pse)}}(T^*))
\tag{5}
$$

Finally, we plug $S'$ into the validation loss and get an approximated objective. Then we update $A$ through the gradient descent:

$$
A \leftarrow A - \eta_a \nabla_A L(S', D^{\text{(val)}})
\tag{6}
$$

where

$$
\begin{aligned}
\nabla_A L(S', D^{\text{(val)}}) &= \frac{\partial T'}{\partial A}\frac{\partial D^{\text{(pse)}}}{\partial T'}\frac{\partial S'}{\partial D^{\text{(pse)}}}\frac{\partial L\left(S', D^{\text{(val)}}\right)}{\partial S'} \\
&\approx \eta_t\eta_s(1-\gamma)\nabla_A(P)\nabla_{T'}(D^{\text{(pse)}})H\nabla_{P,T}^2 L(T, P(A), D_t^{\text{(trn)}})
\end{aligned}
\tag{7}
$$

The second-order term $\nabla_{D^{\text{(pse)}},S}^2 L(S, D^{\text{(pse)}})\nabla_{S'}L(S', D^{\text{(val)}})$ can be approximated by a finite difference Hessian matrix $H = [\nabla_{D^{\text{(pse)}}} L(S^+, D^{\text{(pse)}}) - \nabla_{D^{\text{(pse)}}} L(S^-, D^{\text{(pse)}})]/2\epsilon$. Note that $S^\pm = S \pm \nabla_{S'}L(S', D^{\text{(val)}})$, $\epsilon$ stands for a small number.

## 3.2 DATA-REWEIGHTING GENERATION

Recall that the dataset-weights $P(A)$ are generated with learnable parameters $A$. The proposed framework allows us to provide a generator in various forms. Within the conventional sigmoid activation function $\sigma$, we can use different dataset-weights generators to deal with different types of data noise, especially for long sequence time-series forecasting. Here, we proposed three options:

(a) Identity function, $p_i = \sigma(a_i)$

(b) Normal distribution, $p_i = \sigma\left[\sum_{j=1}^{N/2} \frac{1}{\sqrt{2\pi}a_j} \exp\left(-\frac{(i - a_{j+N/2})^2}{2a_j^2}\right)\right]$ .

(c) Fourier series, $p_i = \sigma\left[\sum_{j=1}^{N/2b} a_j \sin(\frac{2\pi ji}{N}) + \sum_{k=N/(2b+1)}^{N/b} a_k \cos(\frac{2\pi ki}{N})\right]$

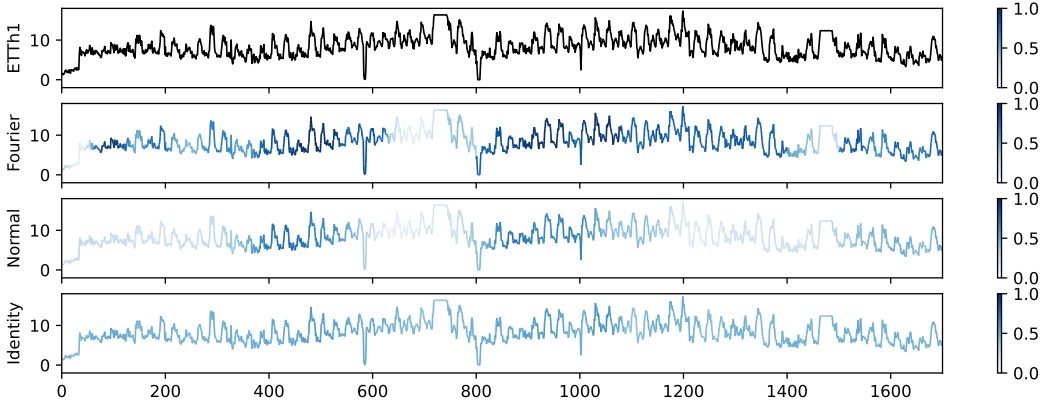

Figure 2: ETTh1 dataset reweighted by the proposed three dataset-weight generators. The lighter colors indicates lower dataset-weights, that is, the targeted noise.

Figure 2 presents a sample of ETTh1 dataset reweighted by the proposed three dataset-weight generators. The line with lighter color indicates that the training data sampled from this area acquire lower weights (from one to zero). We can whiteness two noise areas for long sequence forecasting, around 700-800 and 1400-1500. Among the three methods, the Fourier generator has captured both data outliers with its decomposition in the frequency domain. The normal generator cannot filter outliers in a fine-grained manner, while the identity generator cannot generate highly discriminating weights due to training sparsity sparse training problem, where each parameter in $A$ is trained only once every epoch. For most of the time-series Transformer converging within several epochs, $A$ is unable to be fully trained. After the empirical evaluation in experiments, the last dataset-weights generator becomes the default option.

## 4 CASCADED TEACHING

In this section, we propose a generalized cascaded teaching approach for time-series prediction. There is a sequence of models: $1, \ldots, K$. The first model is a teacher. The $K$-th model is a student. For any other model, it is both a teacher and a student. Model $i$ teaches model $i + 1$. The teaching mechanism is the same as described in Section 3. Given model $i$, we use it to generate a pseudo time-series dataset, then use this pseudo dataset to train model $i + 1$. In our framework, there are $K + 1$ learning stages. For $1 \leq k \leq K$, the $k$-th learning stage corresponds to train the $k$-th model. The $(K + 1)$-th learning stage corresponds to train $A$ via model validation. In the first stage, we train the network $T_1$ (including encoder and decoder) of the first model by solving the following problem:

$$T_1^*(A) = \min_{T_1} L\left(T_1, P(A), D^{(\text{trn})}\right) \qquad , \tag{8}$$

where $A$ contains parameters used to generate dataset-weights in a training set $D^{(\text{trn})}$. In the second stage, model 1 teaches model 2. Given the optimal parameters $T_1^*(A)$ of model 1, we apply it to generate a pseudo-labeled dataset $D^{(\text{pse})}(T_1^*)$ in the same way as described in Section 3. Then we use $D^{(\text{pse})}(T_1^*)$ to train the network parameters $T_2$ of model 2 by solving the following optimization problem:

$$T_2^*(T_1^*(A)) = \min_{T_2} L\left(T_2, D^{(\text{pse})}(T_1^*)\right) \qquad . \tag{9}$$

In the $k$-th stage, model $k - 1$ teaches $k$. Let $T_{k-1}^*(\cdots T_1^*(A))$ denote the optimal parameters of model $k - 1$ trained at stage $k - 1$. We use it to generate a pseudo-labeled dataset $D^{(\text{pse})}(T_{k-1}^*(\cdots T_1^*(A)))$ and use the pseudo-labeled dataset to train the parameters $T_k$ of the model $k$. This amounts to solving the following optimization problem:

$$T_k^* = \min_{T_k} L\left(T_k, D^{(\text{pse})}(T_{k-1}^*(\cdots T_1^*(A)))\right) \qquad . \tag{10}$$

The process continues until all $K$ models are trained. At the $(K+1)$-th stage, we perform validation of these trained models on the validation dataset and learn the architecture $A$ of the first model by

Table 1: Multivariate long sequence time-series forecasting results on two datasets (four cases).

| Methods | | Cas-Informer | | | | Informer | | | Cas-Query-Selector | | | | Query-Selector | |
|---|---|---|---|---|---|---|---|---|---|---|---|---|---|---|
| Role | | Student | | Teacher | | Baseline | | Student | | Teacher | | Baseline | |
| Metric | | MSE | MAE | MSE | MAE | MSE | MAE | MSE | MAE | MSE | MAE | MSE | MAE |
| ETTh₁ 24 | | **0.402** | **0.448** | 0.430 | 0.471 | 0.602 | 0.578 | **0.384** | **0.432** | 0.391 | 0.439 | 0.423 | 0.462 |
| ETTh₁ 48 | | **0.478** | **0.498** | 0.514 | 0.534 | 0.718 | 0.647 | 0.425 | 0.460 | **0.423** | **0.456** | 0.458 | 0.488 |
| ETTh₁ 168 | | **0.768** | **0.681** | 0.784 | 0.768 | 1.001 | 0.797 | **0.641** | **0.588** | 0.651 | 0.591 | 0.684 | 0.609 |
| ETTh₁ 336 | | **0.999** | **0.752** | 1.002 | 0.758 | 1.304 | 0.938 | **0.786** | **0.678** | 0.850 | 0.688 | 0.850 | 0.704 |
| ETTh₁ 720 | | **1.073** | **0.829** | 1.117 | 0.846 | 1.201 | 0.902 | **1.012** | **0.832** | 1.024 | 0.835 | 1.115 | 0.843 |
| ETTh₂ 24 | | 0.613 | 0.626 | **0.609** | **0.611** | 1.792 | 1.078 | 0.408 | 0.491 | **0.391** | **0.481** | 0.412 | 0.486 |
| ETTh₂ 48 | | **0.974** | **0.796** | 1.089 | 0.831 | 2.099 | 1.168 | 0.790 | 0.704 | **0.784** | **0.702** | 1.407 | 0.932 |
| ETTh₂ 168 | | **1.959** | **1.185** | 2.157 | 1.203 | 7.376 | 2.365 | 1.145 | 0.855 | **1.125** | **0.839** | 1.739 | 1.013 |
| ETTh₂ 336 | | **2.195** | **1.223** | 2.197 | 1.229 | 4.466 | 1.779 | 2.195 | 1.075 | **2.154** | **1.040** | 2.317 | 1.186 |
| ETTh₂ 720 | | 2.532 | 1.342 | **2.481** | **1.286** | 3.326 | 1.516 | 2.767 | 1.399 | **2.610** | **1.324** | 3.066 | 1.308 |
| ETTm₁ 24 | | **0.268** | **0.341** | 0.270 | 0.346 | 0.320 | 0.385 | **0.300** | **0.364** | 0.313 | 0.410 | 0.335 | 0.388 |
| ETTm₁ 48 | | 0.389 | 0.440 | **0.378** | **0.426** | 0.461 | 0.466 | 0.440 | 0.458 | **0.440** | **0.453** | 0.473 | 0.470 |
| ETTm₁ 96 | | 0.456 | 0.481 | **0.439** | **0.467** | 0.621 | 0.567 | 0.423 | 0.457 | **0.413** | **0.445** | 0.454 | 0.483 |
| ETTm₁ 288 | | 0.607 | 0.561 | **0.581** | **0.541** | 0.921 | 0.743 | 0.482 | 0.494 | **0.469** | **0.489** | 0.619 | 0.599 |
| ETTm₁ 672 | | 0.648 | 0.592 | **0.626** | **0.572** | 1.045 | 0.790 | 0.602 | 0.569 | **0.593** | **0.563** | 1.127 | 0.841 |
| Weather 24 | | 0.314 | 0.367 | **0.310** | **0.365** | 0.325 | 0.371 | 0.303 | 0.358 | **0.298** | **0.353** | 0.332 | 0.379 |
| Weather 48 | | **0.367** | **0.409** | 0.369 | 0.411 | 0.391 | 0.435 | 0.353 | 0.405 | **0.347** | **0.399** | 0.404 | 0.464 |
| Weather 168 | | 0.496 | 0.506 | **0.472** | **0.489** | 0.568 | 0.557 | 0.487 | 0.504 | **0.479** | **0.498** | 0.601 | 0.562 |
| Weather 336 | | 0.567 | 0.558 | **0.551** | **0.549** | 0.624 | 0.581 | 0.521 | 0.530 | **0.513** | **0.523** | 0.686 | 0.610 |
| Weather 720 | | **0.597** | **0.582** | 0.603 | 0.583 | 0.633 | 0.592 | 0.570 | 0.567 | **0.564** | **0.562** | 0.858 | 0.752 |
| Traffic 96 | | **0.632** | **0.337** | 0.637 | 0.341 | 0.722 | 0.343 | 0.694 | 0.356 | **0.689** | **0.348** | 0.753 | 0.364 |
| Traffic 192 | | **0.623** | **0.334** | 0.633 | 0.346 | 0.708 | 0.383 | 0.702 | 0.361 | **0.694** | **0.356** | 0.763 | 0.381 |
| Traffic 336 | | 0.649 | 0.349 | **0.646** | **0.347** | 0.772 | 0.410 | 0.725 | 0.373 | **0.713** | **0.367** | 0.796 | 0.394 |
| Traffic 720 | | 0.676 | 0.394 | **0.671** | **0.389** | 0.874 | 0.470 | 0.774 | 0.388 | **0.755** | **0.374** | 0.870 | 0.409 |
| Count | | 26 | | 22 | | 0 | | 10 | | 38 | | 0 | |

minimizing the validation loss. The corresponding optimization problem is:

$$\min_A L\left(T_1^*(A), P(A), D^{(\text{val})}\right) + \lambda \sum_{i=2}^{K} L\left(T_i^*, D^{(\text{val})}\right) \qquad . \tag{11}$$

Where $\lambda$ stands for the cascaded teaching rate. Putting these pieces together, we get the following multi-level optimization problem:

$$\begin{aligned}
\min_A \quad & L\left(T_1^*(A), P(A), D^{(\text{val})}\right) + \lambda \sum_{i=2}^{K} L\left(T_i^*, D^{(\text{val})}\right) \\
s.t. \quad & T_K^* = \min_{T_K} L\left(T_K, D^{(\text{pse})}(T_{K-1}^*)\right) \\
& \dots \\
& T_2^* = \min_{T_2} L\left(T_2, D^{(\text{pse})}(T_1^*(A))\right) \\
& T_1^* = \min_{T_1} L\left(T_1, P(A), D^{(\text{trn})}\right)
\end{aligned} \tag{12}$$

From bottom to top, the $K + 1$ optimization problems correspond to the learning stage $1, \dots, K$ respectively. The first $K$ optimization problems are on the constraints of the $(K+1)$-th optimization problem. The $K$ stages are performed end-to-end in a joint manner where different stages mutually influence each other. The detailed optimization algorithm can be found in Appendix.

## 5 EXPERIMENT

We perform experiments on 5 datasets: ETT (ETTh1, ETTh2, ETTm1), WTH and Traffic. Since the framework we proposed only modified the weights on dataset, it can be applied to most time series prediction models under the end-to-end deep learning paradigm. To compare with the SOTA methods, we adopt three Transformer models: Informer Zhou et al. (2021), Query-SelectorKlimek et al. (2021), Reformer Kitaev et al. (2020), and a CNN-based model SCINetLiu et al. (2021). The

last two models are presented in the Appendix C.1 to show the generalizability of the proposed framework. We also present a comprehensive ablation study to test the effectiveness of the different parts of our framework. More details about parameter sensitivity will also be included in Appendix C.3.

## 5.1 DATASETS

**ETT** (Electricity Transformer Temperature) provided by Zhou et al. (2021) records 2-year ETT data from two separated counties in China. The granularity of {ETTh1, ETTh2} and ETTm1 is respectively 1 hour and 15 minutes. Each data point contains 7 channels: 1 oil temperature as target value and 6 power load features.
**WTH** (Weather) contains hourly local climatological data from 2010 to 2013 for around 1600 US locations. Each data point consists of the target value "wet bulb" and 11 other climate features.
**Traffic** is a collection of hourly data from the California Department of Transportation that describes road occupancy measured by different sensors on highways in the San Francisco Bay Area.

## 5.2 EXPERIMENTAL DETAILS

We perform extensive experiments on InformerZhou et al. (2021) and Query-Selector Klimek et al. (2021). **Hyperparameter** By default, the input length of the encoder is set to 96, and the input length of the decoder is set to 48 for Informer. The other hyperparameters are consistent with the two original papers. Due to limited computing resources, the learner number $K$ is set to 2, which follows the canonical teacher-student learning paradigm. The self-study rate $\gamma$ is set to 0.2 so that the student model relies mainly on the quality of the pseudo-labeled dataset. **Platform:** All the models were trained/tested on 2 Nvidia V100 32GB GPU. More information about network setup can be found in the Appendix C.3.

## 5.3 RESULTS AND ANALYSES

The main results are summarized in table 1. There are two comparisons of cascaded teaching's performance on Informer and Query-Selector separately. The winners are in boldface. As we can see in the last row of Table 1, the usage of the cascading framework outperforms the baselines in all datasets, and the improvement is more significant on ETT datasets (improved 66% in $ETTh_2$ at most). For the Informer, the student model performs slightly better than the teacher (Count 26 > 22) The goal of training $A$ helps to reduce the student's validation loss. For the Query-Selector, the student model performs worse than the teacher (Count 10 < 38), but their metrics are very close. The different gap derives from the strong inputs' sparsity assumption in the Query-Selector model because the selected queries have weakened the effect of reweighting. For the fine-grained case ($ETT_{m1}$), the improvement is less obvious than hourly dataset, which indicates that reweighting is more suitable for longer inputs. It is also worth noting that the performance improvement brought by the cascading framework does not decrease significantly with the increase in the prediction length, which reveals its potential for the prediction of longer sequences.

## 5.4 ABLATION STUDY

We design two individual ablation studies to further validate the effectiveness of the different components of the framework: dataset-weight, teacher model and student model.

### 5.4.1 ABLATION STUDY FOR DATASET-WEIGHTS

Unlike conventional teacher-student learning, our framework introduces the data-reweighting mechanism to find valuable time-series samples. Without data-reweighting, the cascaded framework will degenerate into knowledge distillation and perform similarly to the baseline model. We perform experiments on Cas-Informer and KD-Informer (Informer with Knowledge Distillation) Hinton et al. (2015). The KD-Informer simply sets $p_i(A) = 1$ and removes the part of updating $A$ in the optimization algorithm. The experimental results are summarized in Table 2. As we can see, the performance of the model trained by the knowledge distillation framework is consistent with the baseline. We believe the main reason is that the performance of the student model mainly depends

Table 2: Experimental results on ETT dataset with Cas-Informer, KD-Informer and Informer.

| Methods | | Cas-Informer | | KD-Informer | | Informer | |
|---|---|---|---|---|---|---|---|
| Role | | Student | | Student | | Baseline | |
| Metric | | MSE | MAE | MSE | MAE | MSE | MAE |
| ETTh$_1$ 24 | | **0.402** | **0.448** | 0.623 | 0.591 | 0.602 | 0.578 |
| 48 | | **0.478** | **0.498** | 0.732 | 0.655 | 0.718 | 0.647 |
| 168 | | **0.768** | **0.681** | 1.134 | 0.812 | 1.001 | 0.797 |
| 336 | | **0.999** | **0.752** | 1.411 | 0.949 | 1.304 | 0.938 |
| 720 | | **1.073** | **0.829** | 1.317 | 0.926 | 1.201 | 0.902 |
| ETTh$_2$ 24 | | **0.613** | **0.626** | 1.806 | 1.124 | 1.792 | 1.078 |
| 48 | | **0.974** | **0.796** | 2.147 | 1.192 | 2.099 | 1.168 |
| 168 | | **1.959** | **1.185** | 7.625 | 2.528 | 7.376 | 2.365 |
| 336 | | **2.195** | **1.223** | 4.871 | 1.839 | 4.466 | 1.779 |
| 720 | | **2.532** | **1.342** | 3.412 | 1.530 | 3.326 | 1.516 |
| ETTm$_1$ 24 | | **0.268** | **0.341** | 0.312 | 0.375 | 0.320 | 0.385 |
| 48 | | **0.389** | **0.440** | 0.466 | 0.468 | 0.461 | 0.466 |
| 96 | | **0.456** | **0.481** | 0.617 | 0.553 | 0.621 | 0.567 |
| 288 | | **0.607** | **0.561** | 0.942 | 0.762 | 0.921 | 0.743 |
| 672 | | **0.648** | **0.592** | 1.080 | 0.826 | 1.045 | 0.790 |

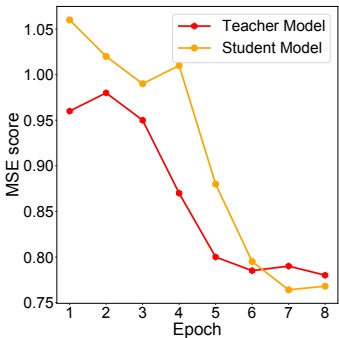

Figure 3: MSE score on test dataset during training.

Table 3: Ablation study of student model's size on ETTh1.

| Cas-informer | | | | | | | | Informer | | | |
|---|---|---|---|---|---|---|---|---|---|---|---|
| Student | | | | Teacher | | | | Baseline | | | |
| Heads | Dim | MSE | MAE | Heads | Dim | MSE | MAE | Heads | Dim | MSE | MAE |
| 8 | 512 | 0.768 | 0.681 | 8 | 512 | 0.784 | 0.690 | 8 | 512 | 0.931 | 0.752 |
| 7 | 448 | 0.803 | 0.702 | 8 | 512 | 0.826 | 0.719 | 7 | 448 | 0.971 | 0.771 |
| 6 | 384 | 1.022 | 0.795 | 8 | 512 | 0.875 | 0.729 | 6 | 384 | 1.063 | 0.817 |
| 5 | 320 | 1.012 | 0.769 | 8 | 512 | 0.874 | 0.738 | 5 | 320 | 1.039 | 0.739 |
| 4 | 256 | 1.037 | 0.808 | 8 | 512 | 0.931 | 0.781 | 4 | 256 | 1.011 | 0.796 |
| 3 | 192 | 0.906 | 0.721 | 8 | 512 | 0.811 | 0.700 | 3 | 192 | 1.041 | 0.814 |
| 2 | 128 | 0.957 | 0.748 | 8 | 512 | 0.852 | 0.733 | 2 | 128 | 1.003 | 0.782 |
| 1 | 64 | 0.967 | 0.751 | 8 | 512 | 0.833 | 0.716 | 1 | 64 | 1.067 | 0.824 |

[1] We have set the prediction length = 168 during experiments.

on the quality of the pseudo label generated by the teacher model, so its MSE score is close to the teacher model. The knowledge distillation framework does not improve the performance of the teacher model. Therefore it has no significant improvement compared to the baseline.

### 5.4.2 ABLATION STUDY FOR TEACHER-STUDENT MODEL

The teacher model generates pseudo labels to help the student learn through the dataset $D^{(pse)}$, whereas the student model updates $A$ by passing the partial gradient to the teacher model. Therefore, the teacher model and student model could have different model sizes. We evaluate the changing of model size by reducing the number of heads and hidden dimensions. Table 3,4 show that the teacher model can still maintain good performance when the size of the student model decreases significantly, while the student model does not exhibit the same performance robustness to the teacher model size. This is because the teacher model can always improve its prediction ability by updating $A$ and $T_1$. Although the gradient quality of A will decrease to a certain extent, there is still a significant improvement compared to the baseline. Conversely, the update of the student model is mainly affected by the quality of pseudo-labeling provided by the teacher model, and thus will have a similar rate of performance degradation as the teacher model.

### 5.5 PARAMETER SENSITIVITY

We perform analysis of parameter sensitivity on 3 hyperparameters of the framework: the weight decay $d_W$, the Fourier divider $b$, and the temperature of the sigmoid function $\mathcal{T}$.

**Weight Decay** In Figure 4(a), we can see that tasks with longer prediction length are more sensitive to the weight decay rate. Therefore a smaller $d_W$ is recommended to achieve stable performance on tasks with different prediction length.

Table 4: Ablation study of teacher model's size on ETTh1.

| Cas-informer | | | | | | | | Informer | | | |
|---|---|---|---|---|---|---|---|---|---|---|---|
| Student | | | | Teacher | | | | Baseline | | | |
| Heads | Dim | MSE | MAE | Heads | Dim | MSE | MAE | Heads | Dim | MSE | MAE |
| 8 | 512 | 0.768 | 0.681 | 8 | 512 | 0.784 | 0.690 | 8 | 512 | 0.931 | 0.752 |
| 8 | 512 | 0.798 | 0.699 | 7 | 448 | 0.848 | 0.731 | 7 | 448 | 0.971 | 0.771 |
| 8 | 512 | 0.872 | 0.722 | 6 | 384 | 0.827 | 0.706 | 6 | 384 | 1.063 | 0.817 |
| 8 | 512 | 0.981 | 0.770 | 5 | 320 | 1.052 | 0.819 | 5 | 320 | 1.039 | 0.739 |
| 8 | 512 | 0.916 | 0.731 | 4 | 256 | 0.965 | 0.760 | 4 | 256 | 1.011 | 0.796 |
| 8 | 512 | 0.961 | 0.763 | 3 | 192 | 0.955 | 0.758 | 3 | 192 | 1.041 | 0.814 |
| 8 | 512 | 0.992 | 0.784 | 2 | 128 | 0.949 | 0.753 | 2 | 128 | 1.003 | 0.782 |
| 8 | 512 | 0.988 | 0.775 | 1 | 64 | 0.974 | 0.765 | 1 | 64 | 1.067 | 0.824 |

[1] We have set the prediction length = 168 during experiments.

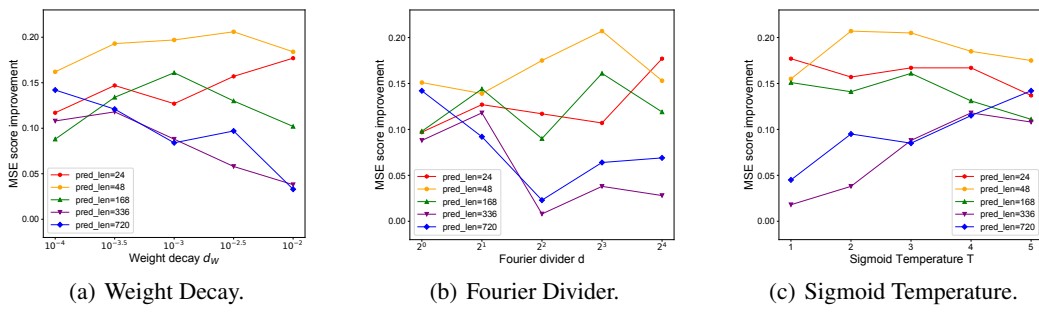

(a) Weight Decay.      (b) Fourier Divider.      (c) Sigmoid Temperature.

Figure 4: The parameter sensitivity of three hyperparameters in Cas-informer.

**Fourier Divider** This hyperparameter controls the amount of parameters in $A$. Figure 4(b) shows that as the prediction task becomes more difficult, the framework requires more parameters of $A$ to control the dataset-weights of each sample at a finer-grained level.

**Sigmoid Temperature** In our paper, we use sigmoid as activation function $\sigma(x) = (1 + e^{-\mathcal{T}x})^{-1}$. A larger sigmoid temperature $\mathcal{T}$ leads to a smoother derivative of the activation function $\sigma$, which allows training of a more selective $A$ for long-sequence predictions.

## 5.6 DISCUSSION

A detailed Analysis is given in Appendix B, which theoretically shows that the dataset weights directly control the magnitude of the student model's weights updating. Therefore, we have the approximated gradient step for the student model $S' = S + g_0 + \sum_{i=1}^{B} p_i g_i$, where $B$ denotes the batch size, and $g_i$ are gradient terms independent of dataset-weights. So the optimization goal becomes: make the student model perform better on the validation dataset by adjusting the update magnitude $p_i$ (i.e. dataset-weights). The experimental results show that the mixture of the reweighted dataset and real-world dataset can even perform better than a single real-world dataset. This is similar to the results in Hwang et al. (2022). The reason for adopting the teacher-student method is that we want to force the teacher to generate more valuable teaching sequences rather than simply finding which sample can make the teacher model have a lower validation loss. The latter may allow the model to perform better on a specific validation set, while the former can aim to improve the model's prediction quality on a different dataset and find samples that better reflect the underlying features of the dataset. The teacher model trained in this way can generalize better on other datasets, where it is the unlabeled dataset $E$ in our case. In addition, this also prevents the model from taking some tricky ways to reduce the loss, such as directly predicting zero for a small absolute value.

## 6 CONCLUSIONS

In this paper, we proposed a cascaded learning framework to reweight the training samples of the teacher model. By using one-step approximation and pseudo dataset, we successfully established a gradient flow between dataset-weights and the test loss of the last model in the framework.

## 7 ETHICS STATEMENT

The proposed cascaded framework can help most time-series forecasting models to have better performance by reweighting the training dataset. This framework can be applied to many significant real-world cases, such as economics and finance forecasting, climate forecasting, disease propagation analysis, traffic management, etc. It also reveals the latent patterns within the dataset, which is inspirational for feature engineering. Our contributions are not limited to LSTF problem. It can also be applied to other tasks like long text generation, anomaly detection, and neural architecture search. In addition, the introduction of teacher-student learning results in the need for more computing resources. In our case, we use two V100 32G GPU in our experiments to achieve a training speed comparable to the baseline model. In addition, the gradient calculation between the two models necessitates distributed training, which increases the difficulty of coding. Therefore, we provide a decoupled distributed code for cascaded learning to facilitate researchers to fill in their own models in our framework.

## 8 REPRODUCIBILITY STATEMENT

We provide an open-source implementation of our cascaded framework at `https://anonymous.4open.science/r/cascaded-framework-6830/`. The detailed hyperparameter setting and analysis can be found in the Appendix.

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

# A    OPTIMIZATION ALGORITHM OF CASCADED TEACHING

In this section, we develop an optimization algorithm to solve the multi-level optimization problem. We first approximate $T_1^*(A)$ using one-step gradient descent w.r.t $L\left(A, T_1, D^{(\text{trn})}\right)$:

$$T_1^*(A) \approx T_1^{'} = T_1 - \eta \nabla_{T_1} L\left(A, T_1, D^{(\text{trn})}\right) \qquad . \tag{13}$$

We substitute $T_1^{'}$ into $L\left(T_2, D^{(\text{pse})}(T_1^*(A))\right)$ and get an approximated objective. Then we approximate $T_2^{'}$ using one-step gradient descent w.r.t the approximated loss as $T_2^* \approx T_2^{'} = T_2 - \eta \nabla_{T_2} L\left(T_2, D^{(\text{pse})}(T_1^{'})\right)$. For the nested function $T_k^*(T_{k-1}^*(\cdots T_1^*(A)))$, we approximate it using one-step gradient descent $T_k^* \approx T_k^{'} = T_k - \eta \nabla_{T_k} L\left(T_k, D^{(\text{pse})}(T_{k-1}^{'})\right)$. In this way, we can plug $\{T_i^{'}\}_{i=1}^{K}$ into the validation loss:

$$L\left(A, T_1^*(A), D^{(\text{val})}\right) + \lambda \sum_{i=2}^{K} L\left(T_i^*, D^{(\text{val})}\right) \qquad .$$

We update $A$ by gradient descend w.r.t the approximated validation loss:

$$A \leftarrow A - \eta \nabla_A \left( L(A, T_1^{'}, D^{(\text{val})}) + \lambda \sum_{i=2}^{K} L(T_i^{'}, D^{(\text{val})}) \right) \qquad . \tag{14}$$

The above derivation makes it possible to generalize our approach in Section 3 to the K-learner case.

# B    DETAILS ANALYSIS

To better understand why our cascaded framework works, we have to use the Lipschitz condition of the transformer-based model. As discussed in Kim et al. (2021), the scaled dot-product self-attention is Lipschitz if the input space is compact. In our case, the compact input space of the time-series dataset can be written as : $[-M, M]^{B \times L \times D}$ where M is the upper bound of the whole dataset and B, L, D represent respectively batch size, input length and hidden dimension. Most of the variants of transformer including Informer and QS-Selector are still continuously differentiable, so the Lipschitz condition applies. Assume that the Lipschitz constant of the cascaded models is K.

In the case of two learners, the teacher model and the student model can be simplified as follows:

$$\begin{aligned} F_T(T, X_i) &= Y_i^T \quad , \\ F_S(S, X_i) &= Y_i^S \quad . \end{aligned} \tag{15}$$

where $F_T$ denotes the teacher model and $F_S$ denotes the student model. For simplicity, we assume that the batch size is 3, $\gamma=0$ and after several iterations the models are close to convergence so that their gradients are a lot smaller than their parameters. Now we do one-step gradient descent on the teacher model in Eq.(4) and use the new parameters to generate pseudo label for a specific sample $e_j$ in the unlabeled dataset $E$. This process can be written as :

$$F_T(T', e_j) = F_T(T + p_1 t_1 + p_2 t_2 + p_3 t_3, e_j) \quad , \tag{16}$$

where $t_i = -\eta_T \frac{\partial}{\partial T} l(T, X_i)$ denotes the parameters update brought by sample $X_i$ . Since $t_i \ll 1$ we can have the following approximation:

$$F_T(T', e_j) = F_T(T, e_j) + (p_1 t_1 + p_2 t_2 + p_3 t_3) \frac{\partial}{\partial T} F_T(T, e_j) \quad . \tag{17}$$

Therefore, when we train student model on the generated pseudo-labeled dataset $D^{(\text{pse})}$, the MSE loss function can be written as :

$$l_j(S, e_j) = (F_S(S, e_j) - F_T(T', e_j))^2$$
$$= (F_S(S, e_j) - F_T(T, e_j) -$$
$$(p_1 t_1 + p_2 t_2 + p_3 t_3) \frac{\partial}{\partial T} F_T(T, e_j))^2 \quad , \tag{18}$$

After omitting the second-order terms, we have:

$$l_j(S, e_j) = l_{j0} + p_1 l_{j1} + p_2 l_{j2} + p_3 l_{j3} \quad ,$$
$$s.t. \quad l_{j0} = (F_S(S, e_j) - F_T(T, e_j))^2 \tag{19}$$
$$l_{ji} = -2t_i \frac{\partial}{\partial T} F_T(T, e_j) (F_S(S, e_j) - F_T(T, e_j))$$

So the one-step gradient descend for student model in Eq.(5) can be rewritten as :

$$S' = S - \eta_S \frac{\partial}{\partial S} l_j(S, e_j)$$
$$= S + g_{j0} + p_1 g_{j1} + p_2 g_{j2} + p_3 g_{j3} \quad ,$$
$$s.t. \quad g_{j0} = -\eta_S \frac{\partial}{\partial S} l_{j0} \tag{20}$$
$$g_{ji} = -\eta_S \frac{\partial}{\partial S} l_{ji}$$

Now we have proven that the dataset-weights directly control the size of the corresponding update of student model parameters.

## C EXTRA EXPERIMENTAL RESULTS

### C.1 SCINET AND REFORMER

In this section, we show the performance of the cascaded framework applied to Reformer Kitaev et al. (2020) and SCINet Liu et al. (2021), a CNN-based neural network for time-series forecasting. The results are summarized in table 5. As we can see, the usage of the cascaded framework helps the two models to have 13% of improvement. This proves the scalability of the cascaded framework on convolutional neural networks and that data-reweighting does not affect the sparse attention mechanism within the Transformer-based models. We also notice that the student SCINet model performs slightly better than the teacher model. Nevertheless, since the self-study rate is set to 0.2, the two models have a very similar improvement compared to the baseline model.

### C.2 DATASET-WEIGHT GENERATOR

We have evaluated the Cas-Informer with different dataset-weight generator. Due to limited time, we could only test normal-distribution-based dataset-weight generator on three datasets, among which the Exchange dataset (financial) records the daily exchange rates of eight different countries ranging from 1990 to 2016. As we can see in table6, the Cas-Informer with Fourier dataset-weight generator perform better on almost all settings except the Exchange dataset. The results confirm the statement about financial datasets in section 3.2 of the paper.

### C.3 HYPERPARAMETERS

The structure of the time-series forecasting model within our cascaded framework is the same as the original setting in Informer Zhou et al. (2021), Query Selector Klimek et al. (2021), Reformer Kitaev et al. (2020) and SCINet Liu et al. (2021). By default, the input length of the encoder is set to 96 and the input length of the decoder is set to 48. We use the Adam optimizer to train for 10 epochs with an initial learning rate of 0.0001 which is halved every 4 epochs. The dataset-weights are also trained by the Adam optimizer with an initial learning rate of 0.0002. Due to GPU memory limitations, we adopt a batch size of 32 and a hidden dimension of 512. Unless otherwise specified, the self-study

Table 5: Multivariate time-series forecasting results on ETT dataset with SCINet and Reformer.

| Methods | | Cas-SCINet | | | | SCINet | | Cas-Reformer | | | | Reformer | |
|---|---|---|---|---|---|---|---|---|---|---|---|---|---|
| Role | | Student | | Teacher | | Baseline | | Student | | Teacher | | Baseline | |
| Metric | | MSE | MAE | MSE | MAE | MSE | MAE | MSE | MAE | MSE | MAE | MSE | MAE |
| ETTh$_1$ 24 | | 0.311 | 0.357 | **0.305** | **0.350** | 0.332 | 0.375 | **0.845** | **0.713** | 0.862 | 0.733 | 0.991 | 0.754 |
| 48 | | **0.360** | **0.392** | 0.367 | 0.398 | 0.408 | 0.430 | **1.123** | **0.835** | 1.158 | 0.840 | 1.313 | 0.906 |
| 168 | | **0.428** | **0.433** | 0.434 | 0.436 | 0.471 | 0.468 | 1.703 | 1.082 | **1.672** | **1.075** | 1.824 | 1.138 |
| 336 | | **0.706** | **0.578** | 0.711 | 0.583 | 0.738 | 0.601 | **1.916** | **1.193** | 1.937 | 1.215 | 2.117 | 1.280 |
| 720 | | **0.717** | **0.584** | 0.722 | 0.589 | 0.761 | 0.628 | 2.163 | 1.512 | **2.156** | **1.501** | 2.415 | 1.520 |
| ETTh$_2$ 24 | | **0.186** | **0.289** | 0.189 | 0.292 | 0.223 | 0.315 | 1.403 | 1.567 | **1.387** | **1.549** | 1.531 | 1.613 |
| 48 | | **0.287** | **0.354** | 0.292 | 0.358 | 0.563 | 0.546 | **1.650** | **1.687** | 1.678 | 1.707 | 1.871 | 1.735 |
| 168 | | **0.502** | **0.489** | 0.519 | 0.494 | 0.586 | 0.543 | **4.254** | **1.513** | 4.289 | 1.545 | 4.660 | 1.846 |
| 336 | | **0.604** | **0.541** | 0.621 | 0.558 | 0.702 | 0.603 | **3.271** | **1.245** | 3.338 | 1.274 | 4.028 | 1.688 |
| 720 | | 1.384 | 0.775 | **1.376** | **0.768** | 1.493 | 0.879 | **4.946** | **1.827** | 4.991 | 1.882 | 5.381 | 2.015 |
| ETTm$_1$ 24 | | **0.238** | **0.301** | 0.242 | 0.302 | 0.277 | 0.329 | 0.668 | 0.591 | **0.652** | **0.576** | 0.724 | 0.607 |
| 48 | | 0.384 | 0.403 | **0.380** | **0.399** | 0.414 | 0.416 | **0.923** | **0.618** | 0.937 | 0.623 | 1.098 | 0.777 |
| 96 | | **0.336** | **0.355** | 0.343 | 0.366 | 0.375 | 0.401 | **1.364** | **0.882** | 1.371 | 0.890 | 1.433 | 0.945 |
| 288 | | **0.502** | **0.526** | 0.513 | 0.535 | 0.582 | 0.556 | **1.615** | **0.965** | 1.648 | 0.982 | 1.820 | 1.094 |
| 672 | | **0.735** | **0.658** | 0.757 | 0.675 | 0.892 | 0.727 | 1.904 | 1.198 | **1.838** | **1.172** | 2.187 | 1.232 |
| Count | | 24 | | 6 | | 0 | | 20 | | 10 | | 0 | |

Table 6: Experimental results on dataset-weight generator.

| Methods | | Cas-Informer(fourier) | | Cas-Informer(normal) | | Informer | |
|---|---|---|---|---|---|---|---|
| Metric | | MSE | MAE | MSE | MAE | MSE | MAE |
| ETTh$_1$ 24 | | **0.473** | **0.492** | 0.526 | 0.524 | 0.577 | 0.549 |
| 48 | | **0.620** | **0.586** | 0.664 | 0.609 | 0.685 | 0.625 |
| 168 | | **0.877** | **0.714** | 0.912 | 0.735 | 0.931 | 0.752 |
| 336 | | **0.957** | **0.725** | 1.077 | 0.821 | 1.128 | 0.873 |
| 720 | | **1.035** | **0.805** | 1.133 | 0.857 | 1.215 | 0.896 |
| ETTh$_2$ 24 | | **0.633** | **0.638** | 0.735 | 0.673 | 0.720 | 0.665 |
| 48 | | **1.146** | **0.875** | 1.418 | 0.988 | 1.457 | 1.001 |
| 168 | | **1.915** | **1.046** | 2.732 | 1.348 | 3.489 | 1.515 |
| 336 | | **2.318** | **1.167** | 2.611 | 1.290 | 2.723 | 1.340 |
| 720 | | **2.479** | **1.336** | 3.124 | 1.422 | 3.467 | 1.473 |
| Exchange 96 | | 0.834 | 0.747 | **0.816** | **0.741** | 0.856 | 0.758 |
| 192 | | 1.009 | 0.802 | **0.975** | **0.788** | 1.221 | 0.905 |
| 336 | | 1.613 | 0.995 | **1.582** | **0.976** | 1.633 | 1.014 |
| 720 | | **2.275** | **1.302** | 2.298 | 1.312 | 2.496 | 1.352 |
| Count | | 22 | | 6 | | 0 | |

rate $\gamma = 0.2$ and the sigmoid temperature $\mathcal{T} = 5$. The early stop algorithm is also adopted to prevent the model from overfitting. We have implemented the cascaded framework in Python 3.8 with Pytorch 1.10 so that the recently released distributed data parallel package can be used.

