# OpenReview forum: "Cascaded Teaching Transformers with Data Reweighting for Long Sequence Time-series Forecasting"
_ICLR.cc/2023/Conference — Submitted to ICLR 2023_

### Official Review · Reviewer_u8Gy · 2022-10-23

**Confidence:** 4
**Clarity, Quality, Novelty And Reproducibility:** See Strength and Weakness.
**Correctness:** 3
**Technical Novelty And Significance:** 2
**Empirical Novelty And Significance:** 2
**Recommendation:** 3

**Strength And Weaknesses:**

Strengths
1. The reweighting idea is reasonable for improving training quality on time series. The learned weights also provide certain interpretation on anomalous data that impair model training.
2. The paper introduces a cascaded teaching framework with sample reweighting.
3. The paper also describes how to generalize the model to many student models and for any base models.

Weakness
1. The overall technical novelty of this paper is not strong. Cascaded knowledge distillation has been proposed before (e.g., Improved Knowledge Distillation via Teacher Assistant, AAAI 2020). Sample reweighing is also a commonly used method for improving training quality.
2. It is the not clear from the paper on why cascading teaching is necessary for the reweighting method. Perhaps knowledge distillation has already performed denoising for certain level. The paper should elaborate on how knowledge distillation and sample reweighting enforce each other and should be combined as an integrated method.
3. In Eq. 2 and 3, it is unclear whether the dataset for training student models ($D^{val}$) contains labels, and whether these datasets have noises. If they have noises, should the training of the student models include another set of reweighting functions?
4. The optimization of the model uses gradient descent, the authors may want to describe the difference between the proposed optimization method and commonly used optimizer (SGD, Adam), and whether the model can be trained by the common optimizers.
5. The proposed framework looks generalizable, and not limited to Transformer. It thus is a little confusing on why to emphasize Transformer in this paper. It is better to demonstrate the generalizable framework by evaluating it with different base models.
6. As for the generalization ability, since the model focuses on the time series forecasting task. It is better to discuss its generalization to other tasks.

**Summary Of The Paper:**

This paper presents a time series reweighting method, integrated with knowledge distillation method, so that noisy input time series play less important roles than others during model training. The reweighting function is parameterized and trained together with the entire teacher-student models via gradient descent. The methods is generalizable and experimented with Transformer for performance evaluation.

**Summary Of The Review:**

The overall novelty of this paper is not strong. The proposed method has several unclear points that may prevent understanding its design. The experiments also remain to be improved to demonstrate generalizing the method to different networks.

---

> ### Author Response · Authors · 2022-11-19
> **Response to Reviewer u8Gy**
>
> We would like to thank reviewers for the constructive feedback, which we will leverage to improve this work. We hope you could revisit our paper (rebuttal version) and reconsider our contributions.
>
> **Q1 Novelty**
>
> Compared with previous cascaded knowledge distillation (CKD) methods, our proposed CKD method has two major novelties. First, our method performs CKD end-to-end in a multi-level optimization-based framework. All models (including teachers, teaching assistants, and students) participating in the CKD process can mutually influence each other. In addition to forwarding KD from teacher to teaching assistant (TA) to the student, our method also facilitates a backward feedback mechanism from student to TA to teacher where the student’s performance guides the learning of TA, and the TA’s performance guides the learning of teacher. In contrast, previous CKD methods perform teacher-to-TA distillation and TA-to-student distillation separately. There is no backward feedback: student’s performance cannot influence TA’s learning, and TA’s performance cannot influence teacher’s learning. Second, the KD mechanism of the method differs from previous methods. Our method performs KD by generating time series data, while previous methods perform KD by predicting pseudo-class labels. Besides, the conventional knowledge distillation mainly focuses on learning a better student learning by using the labels generated by a bigger teacher model, whereas our work mainly focuses on training a better teacher model by forcing it to generate high-quality pseudo-labels with the help of the dataset-weights.
>
> Compared with previous sample reweighting methods which learn sample weights by minimizing a single model’s validation loss in a bi-level optimization based framework, the novelty of our reweighting method is that we learn sample weights by minimizing the validation losses of a sequence of models where cascaded knowledge distillation is performed in a multi-level optimization based framework. Such a cohort of validation losses provides a more robust measurement of sample weights’ correctness: we not only measure whether the sample weights are good for learning the teacher model, but also measure whether they are good for learning teaching assistant and student models. By minimizing the validation losses of a cascade of models instead of a single model, our method can learn sample weights more correctly. In addition, sample reweighting has relatively little research in the area of time series. So our work is an innovation in the field of time series, where sample reweighting is rarely used.
>
> **Q2 Necessity of cascaded learning**
>
> Sorry to make this misunderstanding. As we explained in Section 3 (paragraph after equation 1), the student model is necessary for the training of P(A). Otherwise, a trivial solution will be yielded. Besides, continuous learning is required for the model to adapt to different dataset-weights. Like humans, the interaction between the teacher and the student through pseudo-labels can not only help the student model to learn better, but also prompt the teacher model to generate better labels. Therefore cascading teaching is necessary for sample reweighting.
>
> As we discussed in section 5.4.1 (ablation study of dataset-weights), the improvement of performance is mainly brought by the dataset-weights instead of the knowledge distillation. By removing the dataset-weights, the performance of simple knowledge-distillation-informer is even worse than the baseline. This proves that the knowledge distillation can’t perform denoising alone. The proposed different reweighting ways could be a new paradigram for improving the time-series forecasting performance. Therefore reweighting is an essential part of cascading teaching.

---

> ### Author Response · Authors · 2022-11-19
> **Response to Reviewer u8Gy**
>
> (continue)
>
> **Q3 Dataset for a student model**
>
> Sorry for causing such a misunderstanding. In the paragraph before equation 2, we declare that the training dataset for the student model is a pseudo-labeled dataset generated by the teacher model.
>
> > For each $s_i\in D^{\textrm{(unl)}}$, we use $T^\ast(A)$ to predict the possible outputs $\hat{t_i}$. Then we get a pseudo-labeled time-series dataset $D^{\textrm{(pse)}}(T^\ast(A)) = \{(s_i, \hat{t_i})\}^{L_p}_{i=1}$. We use this pseudo-labeled time-series dataset to train another Transformer as a student model $S$
>
> It is indeed an interesting question about whether we should include another set of reweighting functions. If we introduce a new set of dataset-weights for student weights and update it independantly, a trivial solution will be yielded where the new dataset-weights are all set to zero, just like the teacher model as we explained in Section 3 (paragraph after equation 1). So an alternative solution is to update the dataset-weights of the teacher model and synchronize it to the student model. We have tried this solution, but unfortunately, the experimental results are not satisfactory. We think there are two main reasons: on the one hand, the datasets used by the two models are different, so they cannot share the same set of dataset-weights; on the other hand, sharing P(A) will give the teacher model a dangerous option: directly assign lower dataset-weights to those samples that is badly learned by the student model instead of generating higher quality pseudo-label. Consequently, we only assign dataset-weights on the teacher model, even in the case of multiple cascaded learners.
>
> **Q4 Optimizer**
>
> Sorry to cause such a misunderstanding. We’ve used Adam optimizer to update the model parameters and the dataset-weights. The proposed cascaded framework is not a novel optimizer, but a framework that can compute the gradient by introducing the teacher-student learning and pseudo-labeling. After calculating the gradient (e.g.$\frac{\partial L(S', D^{(val)})}{\partial A}$), we use Adam optimizer to update the parameter, including dataset-weights parameters A and model parameters S and T. Therefore the common optimizers can be used in our framework of the course. We have rephrased the corresponding descriptions.
>
> **Q5 Generalization to other base models**
>
> The experimental results of Cascaded-SCINet can be found in the Appendix (before rebuttal). SCINet is a CNN-based time-series forecasting model that achieved SOTA results on ETT dataset.
>
> Although, as the reviewer said, the cascading framework can be extended to other types of models, we found that it works best on Transformer-based models. We believe the main reason is that the sparse self-attention mechanism itself has a certain ability to exclude outliers. For example, the Informer can omit some values of the attention matrix with the long-tail distribution. Therefore, a sparse Transformer can reduce the training difficulty of dataset-weights.
>
> We supplement experiments on Reformer. The results are summarized in table 1.
>
> Table 1 : Results of Reformer on ETT dataset.
>
> | Predict length | Cas-Reformer Student | Cas-Reformer Teacher | Reformer             |
> | -------------- | -------------------- | -------------------- | -------------------- |
> | ETTh1          |                      |                      |                      |
> | 24             | MSE:0.845 MAE:0.713  | MSE:0.862 MAE:0.733  | MSE:0.991  MAE:0.754 |
> | 48             | MSE:1.123 MAE:0.835  | MSE:1.158 MAE:0.840  | MSE:1.313 MAE:0.906  |
> | 168            | MSE:1.703 MAE:1.082  | MSE:1.672 MAE:1.075  | MSE:1.824 MAE:1.138  |
> | 336            | MSE:1.916 MAE:1.193  | MSE:1.937 MAE:1.215  | MSE:2.117  MAE:1.280 |
> | 720            | MSE:2.163 MAE:1.512  | MSE:2.156 MAE:1.501  | MSE:2.415 MAE:1.520  |
> | ETTh2          |                      |                      |                      |
> | 24             | MSE:0.186  MAE:0.289 | MSE:0.189 MAE:0.292  | MSE:1.531 MAE:1.613  |
> | 48             | MSE:0.287 MAE:0.354  | MSE:0.292 MAE:0.358  | MSE:1.871 MAE:1.735  |
> | 168            | MSE:0.502MAE:0.489   | MSE:0.519 MAE:0.494  | MSE:4.660 MAE:1.846  |
> | 336            | MSE:0.604 MAE:0.541  | MSE:0.621 MAE:0.558  | MSE:4.028 MAE:1.688  |
> | 720            | MSE:1.384 MAE:0.775  | MSE:1.376 MAE:0.768  | MSE:5.381 MAE:2.015  |
> | ETTm1          |                      |                      |                      |
> | 24             | MSE:0.238 MAE:0.301  | MSE:0.242 MAE:0.301  | MSE:0.724 MAE:0.607  |
> | 48             | MSE:0.384 MAE:0.403  | MSE:0.380 MAE:0.399  | MSE:1.098 MAE:0.777  |
> | 96             | MSE:0.336 MAE:0.355  | MSE:0.343 MAE:0.366  | MSE:1.433 MAE:0.945  |
> | 288            | MSE:0.502 MAE:0.526  | MSE:0.513 MAE:0.535  | MSE:1.820 MAE:1.094  |
> | 672            | MSE:0.735 MAE:0.658  | MSE:0.757 MAE:0.675  | MSE:2.187 MAE:1.232  |
> ||

---

> ### Author Response · Authors · 2022-11-19
> **Response to Reviewer u8Gy**
>
> (continue)
>
> **Q6 Generalization to other tasks**
>
> Thanks for your constructive advice. Since the cascade framework requires the teacher model to generate pseudo-labels, it is not suitable for tasks such as time series classification or anomaly detection. We have tested the performance of the cascaded framework on the classification task, and the results show that the dataset-weights trained with pseudo-logit have the problem of gradient disappearance. We will explore the performance of the cascaded framework in other regression tasks in future work.

---

### Official Review · Reviewer_eiCx · 2022-10-24

**Confidence:** 3
**Correctness:** 3
**Technical Novelty And Significance:** 2
**Empirical Novelty And Significance:** 2
**Recommendation:** 5

**Clarity, Quality, Novelty And Reproducibility:**

The presentation is clear and novel. It's a little bit tricky on the reproducibility as it's kind of multi-level training.

**Strength And Weaknesses:**

Strengths:
1. It's novel to propose the teacher-student cascaded teaching Transformers to re-weight samples.
2. The formulation, especially the optimization part is derived in detail.

Weaknesses:
1. The experiments are weak. It's only compared with Informer which is not the state-of-the art method.
2. It's not sure that whether this method is general enough for other Transformer-based models, such as Autoformer, FedFormer and etc.



**Summary Of The Paper:**

This paper proposes a novel learning framework by cascaded teaching Transformers to reweight samples. The framework is formulated as a multi-level optimization and designed with three different dataset-weight generators. It's also novel to use teacher-student framework for time series forecasting problem. The experimental results show the efficacy of proposed method.

**Summary Of The Review:**

This paper proposes a novel learning framework by cascaded teaching Transformers to reweight samples. The framework is formulated as a multi-level optimization and designed with three different dataset-weight generators. The experimental results show the efficacy of proposed method.

However, it doesn't compare with other state-of-the-art methods, such as FedFormer, Autoformer, NHITS and etc. It also didn't show that this method is general enough to extend to other transformer or non-transformer based models.

---

> ### Author Response · Authors · 2022-11-19
> **Response to Reviewer eiCx**
>
> Thanks for your constructive review. We hope you could revisit our paper (rebuttal version) and reconsider our contributions.
>
> Firstly, we would like to emphasize that the main contribution of our work is to mine valuable time series samples within a dataset by using the teacher-student learning method based on pseudo-labeling, and it gives a mathematical derivation that can be generalized to $N$ cascaded learners.
>
> Our framework is mainly aimed at a specific model rather than migrating among multiple models. Therefore, we think Informer, query-selector, and SCINet are representative enough. Among them, SCINet has achieved SOTA results on some settings on the ETT dataset (see the Appendix, before rebuttal). We supplement experiments on Reformer. The results are shown in Table 1.
>
> Table 1 : Results of Reformer on ETT dataset.
>
> | Predict length | Cas-Reformer Student | Cas-Reformer Teacher | Reformer             |
> | -------------- | -------------------- | -------------------- | -------------------- |
> | ETTh1          |                      |                      |                      |
> | 24             | MSE:0.845 MAE:0.713  | MSE:0.862 MAE:0.733  | MSE:0.991  MAE:0.754 |
> | 48             | MSE:1.123 MAE:0.835  | MSE:1.158 MAE:0.840  | MSE:1.313 MAE:0.906  |
> | 168            | MSE:1.703 MAE:1.082  | MSE:1.672 MAE:1.075  | MSE:1.824 MAE:1.138  |
> | 336            | MSE:1.916 MAE:1.193  | MSE:1.937 MAE:1.215  | MSE:2.117  MAE:1.280 |
> | 720            | MSE:2.163 MAE:1.512  | MSE:2.156 MAE:1.501  | MSE:2.415 MAE:1.520  |
> | ETTh2          |                      |                      |                      |
> | 24             | MSE:0.186  MAE:0.289 | MSE:0.189 MAE:0.292  | MSE:1.531 MAE:1.613  |
> | 48             | MSE:0.287 MAE:0.354  | MSE:0.292 MAE:0.358  | MSE:1.871 MAE:1.735  |
> | 168            | MSE:0.502MAE:0.489   | MSE:0.519 MAE:0.494  | MSE:4.660 MAE:1.846  |
> | 336            | MSE:0.604 MAE:0.541  | MSE:0.621 MAE:0.558  | MSE:4.028 MAE:1.688  |
> | 720            | MSE:1.384 MAE:0.775  | MSE:1.376 MAE:0.768  | MSE:5.381 MAE:2.015  |
> | ETTm1          |                      |                      |                      |
> | 24             | MSE:0.238 MAE:0.301  | MSE:0.242 MAE:0.301  | MSE:0.724 MAE:0.607  |
> | 48             | MSE:0.384 MAE:0.403  | MSE:0.380 MAE:0.399  | MSE:1.098 MAE:0.777  |
> | 96             | MSE:0.336 MAE:0.355  | MSE:0.343 MAE:0.366  | MSE:1.433 MAE:0.945  |
> | 288            | MSE:0.502 MAE:0.526  | MSE:0.513 MAE:0.535  | MSE:1.820 MAE:1.094  |
> | 672            | MSE:0.735 MAE:0.658  | MSE:0.757 MAE:0.675  | MSE:2.187 MAE:1.232  |
> ||

---

### Official Review · Reviewer_JAsS · 2022-10-25

**Confidence:** 3
**Correctness:** 3
**Technical Novelty And Significance:** 3
**Empirical Novelty And Significance:** 3
**Recommendation:** 6

**Clarity, Quality, Novelty And Reproducibility:**

Clarity: generally the paper is well-written, with some details unclear to me
 - What is the $x$ in (b) of section 3.2
 - In figure 2, why the weights are applied to different time steps? In my understanding the weights are consistently applied to time series sequences

Novelty: teacher-student model is not novel idea, but its application on dataset reweighting seems new to me
Reproducibility: The paper provides open-sourced implementation

**Strength And Weaknesses:**

Strengths:
+ The paper is well organized and the basic idea is easy to follow
+ The motivation of selecting data samples in training set is convincing.

Weakness
- My main concern is the efficiency  and training stability of this method. The teacher-student model can be difficult to train compared to common supervised methods, and a complexity analysis can be necessary, especially in cascaded teaching
- The Hessian approximation could be also investigated in analysis.
- More state-of-the-art transformers (FEDformer, autoformer, etc) should be investigated


**Summary Of The Paper:**

This paper presents a teacher-student model in long-term time series forecasting. Specifically, a teacher model is learned with reweighted training set.  A student model is then trained on a mixed dataset comprising of the original training set and the pseudo-labeled dataset produced by the teacher. The student is then validated on a separated validation set, where the weights of training samples are trained. The training of both teacher and student only takes one step of gradient descent. This idea can also be extended to a cascaded teaching process.

**Summary Of The Review:**

Overall I think the paper makes reasonable contribution to transformer-based forecasting, though the idea is not quite novel, and I would be happy to see it accepted.

---

> ### Author Response · Authors · 2022-11-19
> **Response to Reviewer JAsS**
>
> We would like to thank the reviewer for the constructive feedback, which we will leverage to improve this work. Here, we will address your concerns one by one.
>
> **Q1 Complexity**
>
> Please refer to our response to Q4 of reviewer 2SH3. The extra runtime is largely due to communication overhead. Our cascaded framework is trained distributedly.
>
> **Q2 Hessian approximation**
>
> As a key technique in one-step gradient descends approximation, the computation of Hessian matrix has already been discussed in the following papers.
>
> [1] Arber Zela, Thomas Elsken, Tonmoy Saikia, Yassine Marrakchi, Thomas Brox, and Frank Hutter. Understanding and robustifying differentiable architecture search. ICLR 2020.
>
> [2] Jonathan Lorraine, Paul Vicol, and David Duvenaud. Optimizing millions of hyperparameters by implicit differentiation. AISTATS 2020. 1540–1552
>
> [3] Dong, Xuanyi, et al. Automated deep learning: Neural architecture search is not the end. arXiv 2021.
>
> [4] Jelena Luketina, Mathias Berglund, Klaus Greff, and Tapani Raiko. Scalable gradient-based tuning of continuous regularization hyperparameters. ICML 2016. 2952–2960.
>
> **Q3 Other transformers**
>
> Our framework is mainly aimed at a specific model rather than transferring among multiple models. Therefore, we think Informer, query-selector, and SCINet are representative enough. Among them, SCINet has achieved SOTA results on several settings on the ETT dataset (see the Appendix, before rebuttal). We perform additional experiments on Reformer. The results are shown in Table 1.
>
> Table 1 : Results of Reformer on ETT dataset.
>
> | Predict length | Cas-Reformer Student | Cas-Reformer Teacher | Reformer             |
> | -------------- | -------------------- | -------------------- | -------------------- |
> | ETTh1          |                      |                      |                      |
> | 24             | MSE:0.845 MAE:0.713  | MSE:0.862 MAE:0.733  | MSE:0.991  MAE:0.754 |
> | 48             | MSE:1.123 MAE:0.835  | MSE:1.158 MAE:0.840  | MSE:1.313 MAE:0.906  |
> | 168            | MSE:1.703 MAE:1.082  | MSE:1.672 MAE:1.075  | MSE:1.824 MAE:1.138  |
> | 336            | MSE:1.916 MAE:1.193  | MSE:1.937 MAE:1.215  | MSE:2.117  MAE:1.280 |
> | 720            | MSE:2.163 MAE:1.512  | MSE:2.156 MAE:1.501  | MSE:2.415 MAE:1.520  |
> | ETTh2          |                      |                      |                      |
> | 24             | MSE:0.186  MAE:0.289 | MSE:0.189 MAE:0.292  | MSE:1.531 MAE:1.613  |
> | 48             | MSE:0.287 MAE:0.354  | MSE:0.292 MAE:0.358  | MSE:1.871 MAE:1.735  |
> | 168            | MSE:0.502MAE:0.489   | MSE:0.519 MAE:0.494  | MSE:4.660 MAE:1.846  |
> | 336            | MSE:0.604 MAE:0.541  | MSE:0.621 MAE:0.558  | MSE:4.028 MAE:1.688  |
> | 720            | MSE:1.384 MAE:0.775  | MSE:1.376 MAE:0.768  | MSE:5.381 MAE:2.015  |
> | ETTm1          |                      |                      |                      |
> | 24             | MSE:0.238 MAE:0.301  | MSE:0.242 MAE:0.301  | MSE:0.724 MAE:0.607  |
> | 48             | MSE:0.384 MAE:0.403  | MSE:0.380 MAE:0.399  | MSE:1.098 MAE:0.777  |
> | 96             | MSE:0.336 MAE:0.355  | MSE:0.343 MAE:0.366  | MSE:1.433 MAE:0.945  |
> | 288            | MSE:0.502 MAE:0.526  | MSE:0.513 MAE:0.535  | MSE:1.820 MAE:1.094  |
> | 672            | MSE:0.735 MAE:0.658  | MSE:0.757 MAE:0.675  | MSE:2.187 MAE:1.232  |
> ||
>
> **Q4 Minor clarity problems**
>
> (1) Thanks for pointing this typo. In Section 3.2 (b), x should be changed to $i$, which is the index of $p_i$.
>
> (2) Sorry for causing such confuse. Please refer to our **response to reviewer 2SH3 Q5 (4)**. The dataset-weights are indeed applied on the whole time-series sample $(s_i, t_i)$.H owever, since there is overlap between different time-series samples, and the samples are shuffled during training, we use the window average method to measure the importance of time points through the dataset-weights of adjacent samples. The importance weights of time point are shown in figure 2 of the paper by shades of color.

---

### Official Review · Reviewer_2SH3 · 2022-10-26

**Confidence:** 5
**Correctness:** 3
**Technical Novelty And Significance:** 2
**Empirical Novelty And Significance:** 2
**Recommendation:** 5

**Clarity, Quality, Novelty And Reproducibility:**

Clarity: The paper is easy to follow for most of the parts, except Section 3. Please see weaknesses point 5.

Quality: Some aspects can be improved, particularly addressing the questions and comments related to Section 3.

Novelty: The idea about using a cascaded teaching framework to learn dataset weights seems novel. However, it introduces a significant computational overhead which needs to be justified in terms of relative performance improvements brought up by properly ablating different additional components involved: (i) sample weights learning (ii) cascaded training of student models on pseudo time-series data. See weaknesses point 1.

Reproducibility: The authors have open sourced the code for reproducing the empirical results.

**Strength And Weaknesses:**

Strengths:

1. This paper focuses on an interesting topic about the re-weighting of a time-series dataset so that outlier and less important samples can be excluded from the training process.

2. The results show generalizability across different datasets and transformer architectures.

Weaknesses:

1. The experiments on different datasets do not cover one important baseline which uses sample reweighting to update the target model i.e. the teacher model, and no student models are involved. This could be formulated as an alternating optimization framework to update the target/teacher model and parameters A (used to compute sample weights). This ablation can essentially help understand the trade-off between the extra computational costs incurred to update a sequence of student transformer models and the performance improvement brought by the feedback of the student models.

2. In the Related Work (Section 2.1), the authors mention that this work focuses on searching the architecture of a teacher model by letting it teach a student model where the student’s architecture is fixed. However, I could not find any results supporting the teacher architecture search in the experiments section. This point should be clarified in the main text.

3. In Section 2.2, the authors mention that the dataset weights trained by the proposed framework are not coupled with the model. Have the authors studied how well the trained dataset weights transfer across different transformer architectures?

4. Comparison among compared baseline methods and proposed framework in terms of computational costs (related to training and pseudo data generation) and running time is missing.

5. Some sections specifically Section 3 can be improved in terms of writing and notational consistencies. Please see the first three points below.


Minor Questions:
1. How is the set of parameters A (used for computing the sample weights) initialized?

2. Clarification about some notations such as $L_A, L_u$, and $L_p$ mentioned on page 3 is missing.

3. The ordering of the inputs to the loss function L() is inconsistent at certain places, for example, Eq (1) and Eq (8).

4. In Figure 2, for the dataset in consideration, how are the input-output pairs $(s_i, t_i)$ generated? To elaborate, given a longer sequence of size > 1600, what stride and window length are used to generate the smaller subsequences $(s_i, t_i)$?

5. Have the authors checked as the training progresses, how well is the pseudo output  (generated by the teacher) correlated with the original training output $(t_i)$ for the same input series $(s_i)$?

6. In Section 5.4.2, what is $W$? Does it denote the teacher model parameters?



**Summary Of The Paper:**

This work proposes a cascaded teaching framework for long-sequence time-series forecasting using transformers. The three-level optimization framework involves updating a sequence of transformer models (teacher and students) and a set of parameters A that are used to compute sample weights. Initially, the teacher model is updated using the re-weighted time-series data. The updated teacher model is then used to generate a pseudo time-series dataset for updating the student model. Finally, the set of parameters A affecting the sample weights is updated based on the performance of the student model on a validation set. The paper empirically validates the proposed framework's effectiveness on five time-series datasets using different transformer architectures and prediction lengths.


**Summary Of The Review:**

The proposed framework involves multiple components, thus a thorough ablation study of additional components is very important.  Currently, the compared methods in the experiments section are not convincing. However, based on the clarifications and answers to the questions in the weaknesses section, I am willing to revise my score.

---

> ### Author Response · Authors · 2022-11-19
> **Response to Reviewer 2SH3**
>
> We would like to thank the reviewer for the constructive feedback, which we will leverage to improve this work. Here, we will address your concerns one by one.
>
> **Q1 ablation study without student model**
>
> We understand that the reviewer recommends us to include the ablation study about the student model. Unfortunately, the updating of parameter A is coupled with the teacher-student learning mechanism. As we explained in Section 3 (paragraph after equation 1), if we update parameter A directly without the student model, a trivial solution will be yielded where all items in P(A) are set to zero.
>
> As for the ablation study for the cascaded training of student models on pseudo time-series data, this is essentially the same as the ablation study for dataset weights. By simply removing the dataset weights ($\forall i, p_i =1)$, this framework degenerates into traditional knowledge distillation, where the student model is trained on the pseudo-labeled dataset in a cascaded manner. Table 2 of the paper shows that without dataset-weights, the performance of KD-Informer is even worse than the baseline (Informer).
>
> **Q2 Architecture search**
>
> Sorry to cause such a misunderstanding. The one-step grandient-descend approximation was derived from Hanxiao Liu’s paper DARTS, where the network architecture is updated. In our paper, the word `architecture` refers to a set of parameters in the broad sense of a model. The dataset-weights parameter A can also be replaced with the genotype architecture parameters proposed in DARTS, which we will explore in our future work. We have already rephrased section 2.1 in the rebuttal version to avoid misunderstandings. Thanks for pointing it out.  This proves that the improvement of the cascaded framework is not brought by the knowledge distillation framework, but by the dataset-weights mechanism.
>
> **Q3 Transfer dataset-weights**
>
> We have performed experiments that transfer query-selector-dataset-weights on Informer. The results are summarized in table 1.
>
> Table 1 Results of Cas-Informer with transferred dataset-weights
>
> | Predict length | Transfered Cas-Informer (Teacher) | Cas-Informer (Teacher) | Baseline Informer   |
> | -------------- | --------------------------------- | ---------------------- | ------------------- |
> | ETTh1          |                                   |                        |                     |
> | 24             | MSE:0.527  MAE:0.523              | MSE:0.482 MAE:0.501    | MSE:0.577 MAE:0.549 |
> | 48             | MSE:0.665 MAE:0.617               | MSE:0.638  MAE:0.603   | MSE:0.685 MAE:0.625 |
> | 168            | MSE:0.901 MAE:0.726               | MSE:0.867 MAE:0.704    | MSE:0.931 MAE:0.752 |
> | 336            | MSE:1.003 MAE:0.769               | MSE:0.982 MAE:0.755    | MSE:1.128 MAE:0.873 |
> | 720            | MSE:1.184 MAE:0.871               | MSE:1.011 MAE:0.793    | MSE:1.215 MAE:0.896 |
> | ETTh2          |                                   |                        |                     |
> | 24             | MSE:0.669  MAE:0.652              | MSE:0.644 MAE:0.649    | MSE:0.720 MAE:0.665 |
> | 48             | MSE:1.277 MAE:0.935               | MSE:1.188 MAE:0.909    | MSE:1.457 MAE:1.001 |
> | 168            | MSE:2.276 MAE:1.143               | MSE:1.937 MAE:1.080    | MSE:3.489 MAE:1.515 |
> | 336            | MSE:2.576 MAE:1.260               | MSE:2.411 MAE:1.217    | MSE:2.723 MAE:1.340 |
> | 720            | MSE:2.825 MAE:1.394               | MSE:2.535 MAE:1.371    | MSE:3.467 MAE:1.473 |
> | ETTm1          |                                   |                        |                     |
> | 24             | MSE:0.317 MAE:0.367               | MSE:0.298 MAE:0.361    | MSE:0.323 MAE:0.369 |
> | 48             | MSE:0.457 MAE:0.479               | MSE:0.455 MAE:0.478    | MSE:0.494 MAE:0.503 |
> | 96             | MSE:447 MAE:0.951                 | MSE:1.401 MAE:0.927    | MSE:1.433 MAE:0.945 |
> | 288            | MSE:1.723 MAE:1.068               | MSE:1.694 MAE:1.046    | MSE:1.820 MAE:1.094 |
> | 672            | MSE:2.018 MAE:1.174               | MSE:1.913 MAE:1.127    | MSE:2.187 MAE:1.232 |
> ||
>
> The results show that the self-trained dataset-weights outperform the transfered dataset-weights. In other words, The choice of outliers for transformer models is different. The outlier of one model may be well handled by the unique attention mechanism of another model. Nevertheless, the transfered Cas-Informer still outperforms the baseline.

---

> ### Author Response · Authors · 2022-11-19
> **Response to Reviewer 2SH3**
>
> (continue)
>
> **Q4 Computational costs**
>
> Under the default setting (batch_size=32, label_length=48, input_length=96, predict_length=48), we’ve compared cas-informer and informer in terms of computational costs on Tesla V100. We want to emphasize that the Cas-Informer runs like six-Informer sequential learning, which arises from the hessian approximation. In our original implementation, different Informer runs on different computation node causing extra communication costs. Meanwhile, the training of dataset-weights is once and for all, that is, the dataset-weights for a specific model obtained through cascaded-framework can be directly used for subsequent training without the additional computational cost.
>
> |                           | Time per batch |
> | ------------------------- | -------------- |
> | Informer                  | 0.001254s      |
> | Cas-Informer              | 0.012502s      |
> | Cas-Informer (standalone) | 0.006078s      |
> ||
>
> **Q5 Minor questions**
>
> (1) In order to avoid introducing prior knowledge, we initialize P(A) to 1, so that all samples are treated equally in the beginning. In the case of the Fourier-dataset-weight-generator, A is initialized to 0 due to the existence of a sigmoid activation function.
>
> $ p_i = \sigma \left[ \sum_{j=1}^{N/2b} a_j \sin (\frac{2\pi_{ji}}{N}) + \sum_{k=N/(2b+1)}^{N/b} a_k \cos (\frac{2\pi_{ki}}{N}) \right] $
>
> This statement will be added to the rebuttal version.
>
> (2) The clarification will be added to section 3.
>
> (3) We will reorder the loss function.
>
> (4) The dataset is generated as follows: given a time-series $[ x_{1}, \dots, x_{m}, \dots, x_{L} ] $ of length $L$, where $x_m$ stands for the $m$-th time point of the series. The $i$-th training sample $(s_i, t_i)$  can be defined as $s_i = (x_i, x_{i+1}, ..., x_{i+l_1-1 }), s_i = (x_{i+l_1}, x_{i+l_1+1}, ..., x_{i+l_1+l_2-1 })$, where $l_1 ,l_2$  stand for the input/output length of the sample. Therefore when we plot figure 2, the importance weight of time point $x_m$ can be defined as $\sum ^m_{i=m-l_1-l_2+1}p_i$.
>
> (5) Thanks to the reweighing mechanism, the teacher model can output high-quality time series predictions. At the input, it is obvious that the outlier is masked as shown in figure 2. However, at the output, the difference between pseudo output and $s_i$ is moderate, hence It is difficult to draw clear rules.
>
> (6)Thanks for your kind correction. We will fix this typo in the rolling version (W is changed to $T_1$ ).

---

### Decision · Program_Chairs · 2023-01-20

**Decision:**

Reject

**Justification For Why Not Higher Score:**

see above

**Justification For Why Not Lower Score:**

see above

**Metareview: Summary, Strengths And Weaknesses:**

Unfortunately, the reviewers were not enthusiastic enough about this paper for it to be considered for acceptance at ICLR 2023. There are just too many papers that reviewers were much more enthusiastic about this year, and ICLR has a very low acceptance rate. The authors are encouraged to take the reviewer comments very seriously, even if there are things you disagree with, and make sure that all issues are addressed, and any potential sources of confusion are completely eliminated, in the next version of the paper. Also, please ensure that these are addressed in the initial paper submission for that next conference.

**Summary Of Ac-Reviewer Meeting:**

see above